🔓 | **Open Peer Review** | Host-Microbial Interactions | Research Article

# Mapping niche-specific two-component system requirements in uropathogenic *Escherichia coli*

John R. Brannon,[1] Seth A. Reasoner,[1] Tomas A. Bermudez,[1] Sarah L. Comer,[1] Michelle A. Wiebe,[1] Taryn L. Dunigan,[1] Connor J. Beebout,[1] Tamia Ross,[1] Adebisi Bamidele,[1] Maria Hadjifrangiskou[1,2]

**ABSTRACT**    Sensory systems allow pathogens to differentiate between different niches and respond to stimuli within them. A major mechanism through which bacteria sense and respond to stimuli in their surroundings is two-component systems (TCSs). TCSs allow for the detection of multiple stimuli to lead to a highly controlled and rapid change in gene expression. Here, we provide a comprehensive list of TCSs important for the pathogenesis of uropathogenic *Escherichia coli* (UPEC). UPEC accounts for >75% of urinary tract infections (UTIs) worldwide. UTIs are most prevalent among people assigned female at birth, with the vagina becoming colonized by UPEC in addition to the gut and the bladder. In the bladder, adherence to the urothelium triggers *E. coli* invasion of bladder cells and an intracellular pathogenic cascade. Intracellular *E. coli* are safely hidden from host neutrophils, competition from the microbiota, and antibiotics that kill extracellular *E. coli*. To survive in these intimately connected, yet physiologically diverse niches *E. coli* must rapidly coordinate metabolic and virulence systems in response to the distinct stimuli encountered in each environment. We hypothesized that specific TCSs allow UPEC to sense these diverse environments encountered during infection with built-in redundant safeguards. Here, we created a library of isogenic TCS deletion mutants that we leveraged to map distinct TCS contributions to infection. We identify—for the first time—a comprehensive panel of UPEC TCSs that are critical for infection of the genitourinary tract and report that the TCSs mediating colonization of the bladder, kidneys, or vagina are distinct.

**IMPORTANCE**    While two-component system (TCS) signaling has been investigated at depth in model strains of *Escherichia coli*, there have been no studies to elucidate—at a systems level—which TCSs are important during infection by pathogenic *Escherichia coli*. Here, we report the generation of a markerless TCS deletion library in a uropathogenic *E. coli* (UPEC) isolate that can be leveraged for dissecting the role of TCS signaling in different aspects of pathogenesis. We use this library to demonstrate, for the first time in UPEC, that niche-specific colonization is guided by distinct TCS groups.

**KEYWORDS**    UTI, UPEC, two-component systems, signal transduction

Address correspondence to Maria Hadjifrangiskou, maria.hadjifrangiskou@vumc.org.

John R. Brannon, Seth A. Reasoner, and Tomas A. Bermudez contributed equally to this article. The author order was determined based on contribution of each person.

The authors declare no conflict of interest.

See the funding table on p. 13.

**B**acteria efficiently adapt to environmental changes such as newly colonized niches by rapidly changing gene expression (1–3). A major mechanism through which bacteria interpret environmental changes into specific changes in their gene expression is two-component signal transduction systems (TCSs). TCSs usually comprised two parts: a sensor histidine kinase (HK) and a cognate response regulator (RR). Typically, signal interception results in HK autophosphorylation at a conserved histidine residue and subsequent phosphotransfer to a conserved aspartate on the RR. The most widely observed outcome of RR phosphorylation is increased DNA binding affinity of the

phosphorylated RR for its target promoters, consequently altering target gene expression.

TCSs serve as compartmentalized bacterial logic gates that process sensory input with the net output result often being a change in gene expression that ultimately changes one or multiple bacterial phenotypes. However, multiple studies show that—like in mammalian signaling systems—TCSs are dynamic and branch to incorporate multiple stimuli, interact outside the boundaries of cognate partners, or be part of phosphorelays that allow for a refined, beneficial orchestration of molecular systems (2). While these phenomena are well characterized *in vitro*, either in biochemical protein-protein interactions or in the test tube, few studies have investigated how each TCS of a given pathogen contributes to host colonization.

Uropathogenic *Escherichia coli* (UPEC) is the causative agent of ~80% of the 150 million urinary tract infections (UTIs) that occur annually (4–6). UPEC's ability to survive within several different environments contributes to its successful prevalence. Within the intestinal tract, UPEC colonizes the human gut alongside commensals for extended periods of time, in contrast to diarrheagenic *E. coli* pathotypes (6–8). However, unlike commensals, the genetically diverse UPEC strains are equipped with fitness determinants that allow them to expand beyond the gut (8–11). Exit from the gut is followed by urethral accession to the bladder causing cystitis. During bladder infection, UPEC dynamically reaches the kidney and in some cases can cause pyelonephritis, from where bacteria can traverse to the bloodstream, leading to bacteremia (12–14). In people assigned female at birth, who are disproportionally impacted by UTIs, UTI pathogenesis additionally encompasses the vagina (15, 16). Within these different host niches, UPEC is found in extracellular and intracellular compartments exposed accordingly to different stresses and metabolite inventories. In the bladder and vaginal lumens, UPEC exists planktonically or associated with the urothelial or vaginal cells. Likewise in the kidney, UPEC associates with the host cell membrane (17). In the host intracellular environments, UPEC forms three different intracellular bacterial reservoirs: metabolically active, biofilm-like, intracellular bacterial communities (IBCs) in the superficial umbrella cells of the bladder; quiescent intracellular reservoirs inside transitional bladder cells; and vaginal intracellular communities within vaginal epithelial cells (15, 18).

While the field has developed an in-depth understanding of the molecular systems contributing to UPEC's plasticity, such as flagella curli, and type 1 and P pili, little is known about which TCSs are coordinately used during infection. In this work, we tested the hypothesis that the different unique niches UPEC encounters necessitate the use of distinct TCSs. To test this hypothesis, we constructed an isogenic deletion mutant library of all the TCSs encoded by the prototypical cystitis isolate UTI89 (Table 1) that is extensively used in the field to study UTI pathogenesis (19). As this is the first time, a comprehensive TCS deletion library has been constructed for study in a UPEC isolate, an in-depth *in vitro* analysis of how each TCS deletion strain associates with bladder cells was first performed. Next, we leveraged our well-established UTI mouse models to determine the pathogenicity of each TCS deletion mutant in the bladder, kidneys, and vagina, following transurethral inoculation. We report that *in vitro*, none of the TCS deletions substantially influences adherence or invasion of urothelial cells, indicating that this critical step in infection establishment is under redundant control. Our *in vivo* data uncover—for the first time—the inventory of TCSs needed for bacterial expansion following the adherence and invasion steps in the bladder. These TCSs are distinct from the TCSs that we found to be critical for kidney or vaginal colonization following bladder infection. Collectively, our results demonstrate niche-specific requirements for TCSs during the different stages of UTIs and lay the groundwork for delineating mechanistic details associated with each network during UPEC infection.

**TABLE 1** Information on HK–RR pairs and orphan proteins in UPEC strain UTI89[a]

| HK | Loci | RR | Loci | Ligand or stimulating conditions | Notes | Ref. |
|---|---|---|---|---|---|---|
| ArcB | C3646 | ArcA | C5174 | Anaerobic growth | | (20, 21) |
| AtoS | C2501 | AtoC | C2502 | Acetoacetate | | (22) |
| BaeS | C2353 | BaeR | C2354 | Envelope stress, indole, and flavonoids | | (23) |
| BarA | C3156 | UvrY | C2115 | Acetate and short-chain fatty acids | Deleted separately | (24) |
| BtsS | C2398 | BtsR | C2397 | Pyruvate | | (25) |
| | C4932 | | C4931 | Unknown | | |
| | C4937 | | | Unknown | | |
| CpxA | C4495 | CpxR | C4496 | Envelope stress | | (26) |
| CreC | C5170 | CreB | C5171 | During glucose fermentation | | (27) |
| CusS | C0570 | CusR | C0571 | Copper and silver | | (28) |
| DcuS | C4719 | DcuR | C4718 | C4-dicarboxylates like malate | | (29) |
| DpiB | C0624 | DpiA | C0624 | Citrate | | (30) |
| EnvZ | C3904 | OmpR | C3905 | Changes in osmolarity | | (31) |
| EvgS | C2702 | EvgA | C2701 | Alkali metals and low pH | | (32) |
| GlnL | C4458 | GlnG | C4457 | Nitrogen deprivation | | (33) |
| KdpD | C0699 | KdpE | C0698 | Potassium limitation | | (34) |
| KguS | C4639 | KguR | C4638 | α-Ketoglutarate utilization | | (35, 36) |
| NarQ | C2795 | | | Nitrate and nitrite | | (37) |
| NarX | C1418 | NarL | C1417 | Nitrate and nitrite | | (37, 38) |
| PhoQ | C1258 | PhoP | C1259 | Low magnesium and calcium and membrane stress | | (39, 40) |
| PhoR | C0421 | PhoB | C0420 | Inorganic phosphate | | (41) |
| PmrB | C4706 | PmrA | C4707 | Ferric iron, membrane stress | | (42) |
| QseC | C3451 | QseB | C3450 | Unknown in UPEC | | (43) |
| QseE | C2876 | QseF | C2873 | Epinephrine and norepinephrine in EHEC | Deleted separately | (44) |
| RcsCD | C2500, C2498 | RcsB | C2499 | Osmotic shock | RcsD and RcsB were deleted | (45) |
| RstB | C1796 | RstA | C1797 | Unknown | | |
| TorS | C1056 | TorR | C1059 | Trimethylamine-N-oxide | | (46) |
| UhpB | C4224 | UhpA | C4225 | Glucose-6-phosphate | | (47) |
| YedV | C2167 | YedW | C2168 | Hydrogen peroxide responsive | | (48) |
| YpdA | C2712 | YpdB | C2713 | High pyruvate concentrations | | (25) |
| ZraS | C3816 | ZraR | C3815 | Zinc | | (49) |
| | | NarP | C2471 | Nitrate and nitrite | | (37) |
| | | RssB | C1431 | Glucose, phosphate, nitrogen deprivation | | (50) |

[a]All information was compiled using P2CS and MiST3.0 databases.

## MATERIALS AND METHODS

### Growth of eukaryotic cells and bacterial strains

A complete list of *E. coli* strains and their characteristics used for this study are listed in Table S1. From freezer stocks, strains were grown in 5 mL of lysogeny broth (LB) at 37°C with shaking. For bacteriological assays, LB culture tubes were incubated overnight. For inoculation of 5637 (ATCC HTB-9) cells and C3H/HeN mice, bacterial cultures were first seeded from a freezer stock and incubated for 4 h at 37°C, followed by two sequential, ~24 h sub-cultures (1:1,000 dilution) in 10 mL, static LB media at 37°C in order to induce type 1 pili (19). Preceding inoculation of immortalized cells and mice, strains were normalized in 1× PBS (phosphate-buffered saline). The immortalized bladder epithelial cell line 5637 (ATCC HTB-9) was grown statically at 37°C in RPMI 1640 (Life Technologies Co., Grand Island, NY, USA) media supplemented with 10% FBS (fetal bovine serum) under 5% $CO_2$.

## Construction of UPEC TCS deletion library

A complete list of plasmids and primers can be found in Tables S1 and S2, respectively. For our study, we used the genetically tractable, cystitis strain UTI89 and using the prokaryotic TCS databases, P2CS and MiST, identified a comprehensive list of TCSs in UTI89 for deletion (51, 52). Targeted TCS deletion mutants for the isogenic deletion library were constructed using the λ Red recombinase system, and gene deletion was confirmed by PCR with test primers (53–55).

## Bacterial growth curves

Strains were grown overnight in LB broth and sub-cultured to a starting $OD_{600}$ (optical density at 600 nm) of 0.05 in either LB broth, N-minimal media, or pooled filter-sterilized human urine. Urine was collected under IRB 151465, pooled from multiple healthy donors, and sterilized through a 0.22-µm filter. N-minimal media were prepared as previously described with final composition of 5 mM KCl, 7.5 mM $(NH_4)_2SO_4$, 0.5 mM $K_2SO_4$, 1 mM $KH_2PO_4$, 0.1 mM Tris-HCl (pH 7.4), 10 mM $MgCl_2$, 0.2% glucose, 0.1% casamino acids, 38 mM glycerol, and 10 µg/mL niacin (42, 56, 57). Following inoculation in the specified media, 96-well plates were incubated at 37°C with shaking for 8 h, and $OD_{600}$ readings were taken every 15 min in a SpectraMax i3x (Molecular Devices) plate reader. Calculation of $\mu$ (growth rate) was performed using the formula: $\ln(OD_{600}$ at T2$)$ $- \ln(OD_{600}$ at T1$)$/T2 $-$ T1, where T1 marks the beginning of exponential phase and T2 marks the end of exponential phase (Raw Data in Supplementary Source Data File 1). Division time was calculated using the formula $\ln(2)/\mu$.

## Gentamicin-based protection assays

Experiments were performed as described elsewhere (58). 5637 (ATCC HTB-9) bladder epithelial cells were grown to at least 90% confluency in 24-well plates. Prior to inoculation with *E. coli*, new RPMI 1640 media supplemented with 10% FBS was added to the cells. To achieve an approximate multiplicity of infection (MOI) of 5, 5637 cell density was enumerated to determine the amount of *E. coli* suspension to be used per well. Strain inoculum was added to three sets of triplicate wells and centrifuged for 5 min at $600 \times g$ to facilitate uniform contact between bladder and bacteria cells. Following, plates were incubated at 5% $CO_2$ and 37°C for 2 h. For cell lysis, a final concentration of 0.1% Triton X-100 was used. Bacterial burden was enumerated by serial dilution and spot platting onto LB plates. One set of wells was lysed to determine the total number of bacteria within the well. The other two sets were washed with 0.5 mL of PBS three times. The *E. coli* adhering to bladder cells was enumerated in a set of wells that was immediately lysed after the washes. The final set of wells was gently washed with PBS with 100 µg/mL of gentamicin (Life Technologies Co., Grand Island, NY, USA) for 2 h; afterward, wells were washed two more times with 1 mL of PBS to enumerate the intracellular *E. coli*. The percentage of *E. coli* adherence and invasion were calculated as a percentage of the total number of bacteria.

## Animal studies

All animal experiments were performed using approved protocols in VUMC Institutional Animal Care and Use Committee (IACUC) protocol number M1800101-01. Mice were infected via transurethral inoculation as previously described (59, 60). To prepare the inocula, *E. coli* strains were incubated at 37°C, initially in a 5-mL LB culture tube with shaking for 4 h and followed by two sequential sub-cultures at 1:1,000 into 10 mL of fresh LB and grown statically for 24 h. C3H/HeN female mice aged 7–8 weeks were transurethrally inoculated with 50-µL *E. coli* suspension of $10^7$ CFUs in PBS, and mice were humanely euthanized at 6 h, 24 h, or 28 days post-infection (h/dpi). After euthanasia, organs were removed and homogenized in PBS for CFU enumeration. For quantification of intracellular bacteria, bladder tissue was bisected, incubated in 100 µg/mL of gentamicin for 2 h, washed with PBS, and homogenized in PBS with 0.1% Triton X-100. To

evaluate the long-term fitness of each mutant, separate cohorts of mice were transurethrally inoculated with each strain, and chronic colonization was tracked using longitudinal urinalysis. Specifically, urine was collected for each mouse at 1, 3, 7, 14, 21, and 27 dpi. If urine titers for an individual mouse reached below $10^4$ CFUs/mL, that mouse was considered as resolving bacteriuria and separated to a clean cage to avoid re-infection by the chronically colonized cage-mates (60). Data are presented as time-to-resolution of persistent bacteriuria (Fig. 3D), wherein resolution is a one-time event regardless of whether the mouse experiences high-titer bacteriuria again. Urine CFU data collected over time are presented in Fig. S2, and raw data can be found in Supplementary Source Data File 2.

## Visualization and enumeration of intracellular bacterial communities

IBC enumeration was performed as described previously, using bacterial strains transformed with pCOM::GFP plasmid (13). Mice were euthanized at 6 hpi, and the bladders were removed with aseptic technique. Mouse bladders were stretched, pinned, and fixed with 3.4% paraformaldehyde overnight at 4°C. Bladders were washed twice in 1× PBS, permeabilized with 0.1% Triton X-100 for 15 min, followed by a 1× PBS wash. Bladders were stained at room temperature with Alexa Fluor 568 Phalloidin (Thermo-Fisher) and mounted onto slides with ProLong Diamond Antifade (ThermoFisher). IBCs were manually counted via fluorescence microscopy on an LSM 710 confocal laser scanning microscope (Zeiss).

## Statistical analysis

Statistical analyses were performed in GraphPad Prism software, using the most appropriate test for each analysis. Experiments were performed in accordance with the standard convention, incorporating at least three biological replicates. Statistical tests used for analysis are two-tailed. Additionally, for mouse infections, power analyses were performed and determined that seven subjects per group were needed to achieve a power level of 90% for detecting a 25% difference in the CFU means with an in-group standard deviation of 20%. Additional experimental details of group size, statistical test, error bars, and probability value for each statistically evaluated experiment are specified in the corresponding figures, legends, and text.

## RESULTS

### Construction of a UPEC TCS deletion library

Using the P2CS and MiST3.0 databases, we compiled a list of TCS within the UPEC strain UTI89. In UTI89, we found 32 RRs and HKs, including four hybrid HKs (Table 1). Typically, classical TCSs components are encoded together in an operon; however, in some cases, TCS genes are encoded at distinct loci in the chromosome or comprise more complex multi-branch systems or phosphorelay systems, such as RcsCDB. Finally, certain strains, including UTI89, may encode "orphan" TCS components with no known interaction partners. Within UTI89, we identified 25 orthodox pairs and 14 orphans, including one without a known partner (Table 1). In order to construct a comprehensive TCS deletion library in UTI89 accounting for gene separation, we generated 38 different isogenic deletion mutants. Orthodox TCSs and those encoded in operons were deleted in pairs, while orphans or non-operonic TCS pairs were deleted separately. The *rcsDB* gene cluster, which codes for the phosphorelay system RcsCDB, was deleted as a unit. The CheA-CheY chemosensory system, which has been heavily investigated in the context of chemotaxis (61, 62), was omitted from this study. All resulting mutant strains are marker-less and validated by PCR. To our knowledge, this is the first time a TCS deletion library has been constructed in a UPEC isolate.

Prior to evaluating the contribution of each TCS to UPEC pathogenesis, we sought to determine whether the deletion of any TCS negatively affects UPEC growth *in vitro*. For these studies, we evaluated growth in standard nutrient-rich [lysogeny broth (LB)]

or nutrient-poor (N-minimal) (42) laboratory media commonly used in research labs worldwide. Additionally, given our goal to understand TCS contribution to UTI pathogenesis, we also evaluated the growth of the TCS mutants in pooled human urine.

To manage the growth of so many strains, growth curves were performed in a SpectraMax plate reader, started at an $OD_{600}$ of 0.05 from overnight cultures in LB. As a result of smaller growth volumes and less aeration, the specific growth rate tends to be slower, compared to during growth in batch culture, with a maximal $OD_{600}$ of 0.90–1.0 in LB for wild-type UTI89 (Fig. 1D), compared to an $OD_{600}$ of 4–6 during growth in batch culture.

We observed a shorter division time in strains deleted for the histidine kinase *qseE* in LB (Fig. 1A), and *cpxRA*, *creBC*, *envZompR*, and *qseE* in N-minimal media (Fig. 1B). Accelerated growth with deletion of a TCS is an intriguing observation and forms the basis for future investigations of the nutritional requirements and metabolic propensities of these TCS deletion strains. Under these three specific growth conditions, we observed the highest variability during growth in urine (Fig. 1C), which could be attributed to the fact that this is pooled donated human urine and therefore, nutritionally undefined. Comparison of the TCS mutants during growth in urine revealed no significant differences in doubling time (Fig. 1C). Taken together, these results indicate that—under the conditions tested—deletion of a single TCS is not sufficient to impart a planktonic growth defect.

## Adherence and invasion of urothelial cells are not impaired in UPEC TCS mutants

UPEC infection begins with adherence and invasion of urothelial cells (63). This process is governed by type 1 pili (*fim*), which are adhesive fibers assembled by the chaperone-usher pathway (64). Adherence is key to the ability of UPEC to prevent being eliminated from the urinary tract via urination. Urination is, in fact, the first bottleneck bacteria encounter in the bladder and is responsible for the elimination of ~90% of the cohort, with ~10% of bacteria adhering to the urothelium (58). Variable production of pili per bacterial cell also influences adherence (65). Type 1 pili-mediated adherence triggers internalization of UPEC by the urothelial cell (58, 66). To determine if any of the TCS mutants display adherence or invasion defects, we leveraged a well-established tissue culture model (58) using the 5637 immortalized urothelial cell line. These assays revealed that overall, none of the TCS mutants had significantly different adherence or internalization titers compared to wild-type UTI89 (Fig. 2). These data again indicate that—at least *in vitro* and under the conditions tested—no single TCS deletion impairs the initial steps in UPEC pathogenesis.

## Distinct TCSs contribute to colonization of genitourinary tract niches

We next sought to evaluate the contribution of each TCS mutant in a UTI mouse model. In this model, bacteria are instilled in the bladder via transurethral inoculation, and mouse infection is monitored by CFU of organs and urine (15, 59, 67). Acute infection hallmarks include the formation of intracellular bacterial communities (IBCs) at 6 hpi (59, 68) and bacteriuria that persists over time in ~50% of the infected mice (60). Another hallmark of UTI that is captured in this murine model is the formation of asymptomatic reservoirs in the vagina (15).

In this experiment, we asked how each TCS mutant colonizes the bladder, kidneys, and vagina, following transurethral inoculation. Cohorts of 6–8-week-old female C3H/HeN mice were thus transurethrally inoculated with the wild-type parent or each of the isogenic TCS mutants. Mice were euthanized at 24 hpi, and bacterial titers in the bladder, kidneys, and vagina were enumerated for each infected mouse. Consistent with the hypothesis that distinct TCSs are needed in unique sub-niches in the genitourinary tract, we observed mutants with niche-specific defects (Fig. 3).

In the bladder, the Δ*arcA*Δ*arcB*, Δ*narXL*, Δ*phoPQ*, and D*rssB* showed decreased titers compared to the parent strain at 24 hpi (Fig. 3A) but were able to colonize the kidney and

**A)** **Lysogeny Broth**

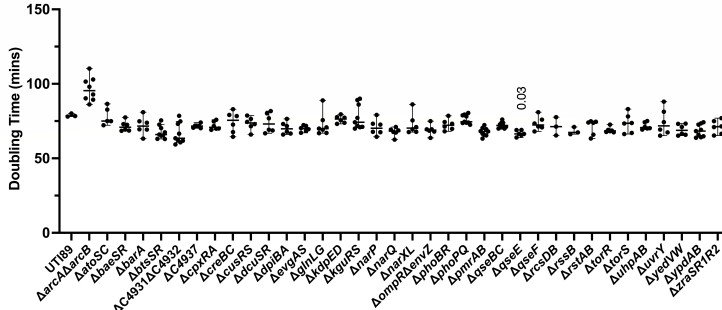

**B)** **N-Minimal Media**

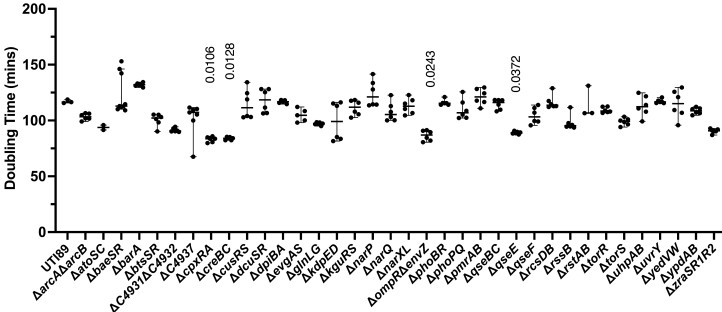

**C)** **Pooled Human Urine**

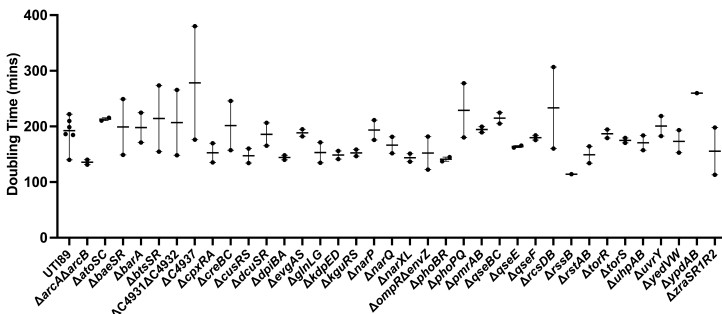

**D)** **Maximum OD$_{600}$**

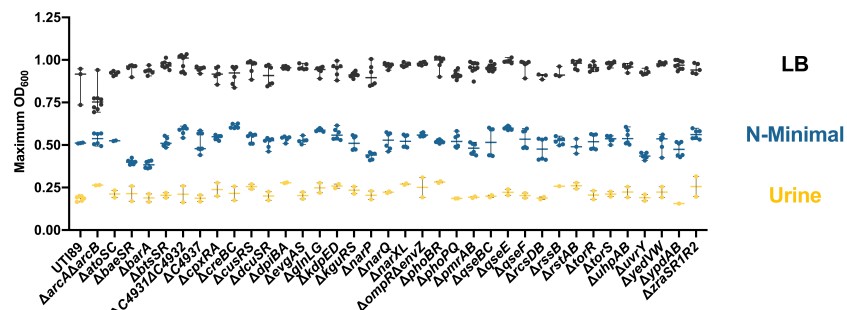

**FIG 1** Growth of UPEC TCS deletion mutants. (A–C) Graphs depict the doubling time (minutes) of each TCS mutant compared to the isogenic parent UTI89, during growth in (A) LB media, (B) N-minimal media, or (C) pooled human urine with shaking, at 37°C in a SpectraMax i3x (Molecular Devices) plate reader. Doubling time was calculated as described in the Materials and Methods. Horizontal line indicates the median, and error bars depict the 95% confidence interval. A non-parametric Kruskal–Wallis with two-sided Dunn's *post hoc* test was performed for statistical analysis. (D) Maximum OD$_{600}$ from growth curves conducted in LB, N-minimal media, or pooled human urine. Error bars depict median and 95% confidence interval.

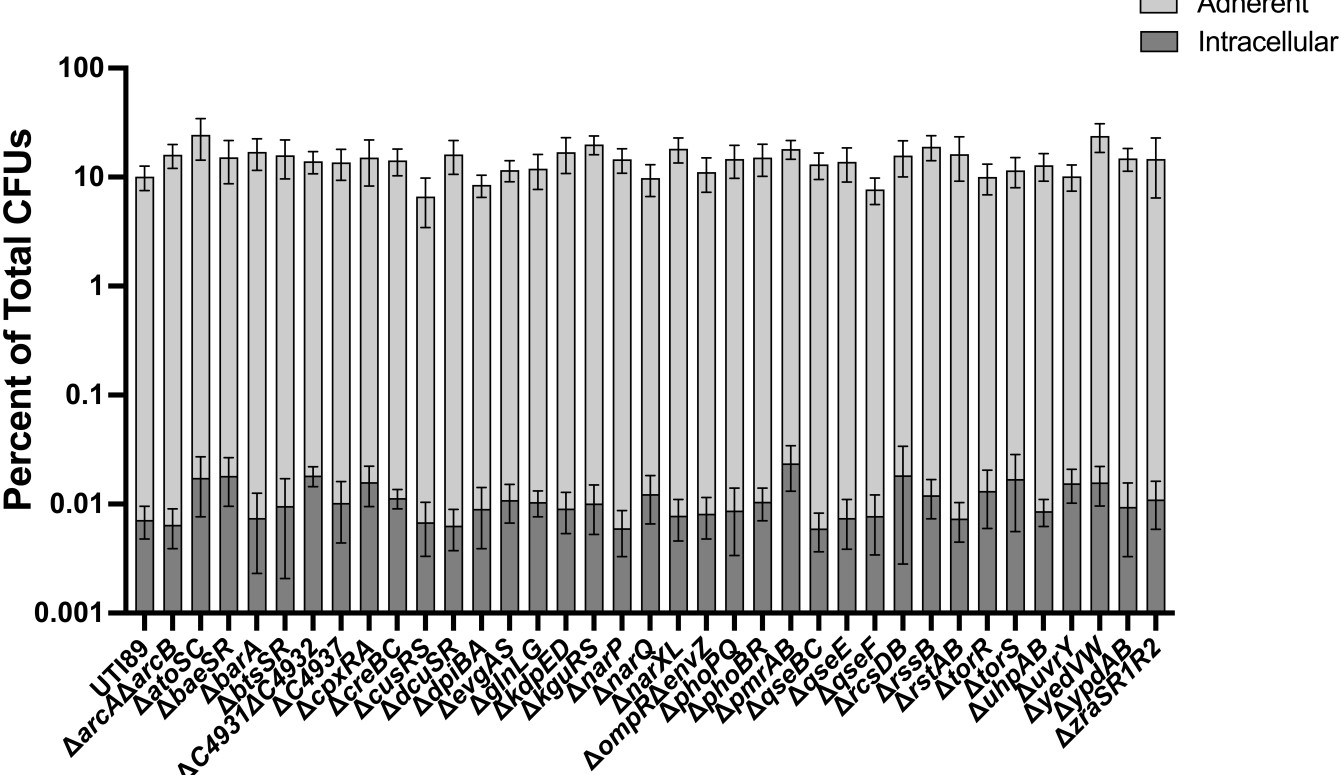

**FIG 2** Adherence and invasion of UPEC TCS deletion mutants to bladder epithelial cells. Adherent and intracellular bacterial titers for wild-type UTI89 and each of the isogenic TCS deletion strains. Experiments were performed using an MOI of 5 on the immortalized urothelial cell line 5637. The percentage of *E. coli* adherence and invasion were calculated as a percentage of the total number of bacteria with a well at the 2 h endpoint. Each bar represents the mean, and error bars depict the 95% confidence interval. A non-parametric Kruskal–Wallis with two-sided Dunn's *post hoc* test was performed for statistical analysis.

transit to the vagina similar to the wild-type strain (Fig. 3B and C). In the kidney, ΔC4937, Δ*cpxRA*, Δ*cusRS*, and Δ*rstAB* exhibited decreased titers compared to the wild-type parent (Fig. 3B), while Δ*torS*, Δ*creBC*, and Δ*glnLG* exhibited significant colonization defects in the vagina (Fig. 3C). Notably, the uncharacterized system ΔC4931ΔC4932 displayed an apparent growth advantage in the kidney niche. These observations indicate niche-specific responses to the local micro-environment that necessitate the use of distinct TCS.

## Mutants defective for bladder colonization are associated with energy metabolism

Our lab has previously elucidated that aerobic respiration is critical for intracellular replication of UPEC (69, 70). The current analyses uncover four TCS mutants, Δ*arcA*Δ*arcB*, Δ*narXL*, Δ*phoPQ*, and the Δ*rssB* RR that display defects in bladder colonization at 24 hpi. ArcA/ArcB, NarX/NarL, and RssB belong to regulatory networks associated with respiration (71, 72), while PhoP/PhoQ is implicated in stress response and energy metabolism (73, 74). We, therefore, focused on these regulators to further dissect the stage at which they become important during infection and to evaluate how their deletion impacts long-term persistence of UPEC in the urinary tract.

During UTI, UPEC becomes internalized by urothelial cells, in which they replicate into biofilm-like communities by consuming oxygen primarily via the quinol oxidase cytochrome bd (70). Previous work demonstrated that the activity of the ArcB HK is influenced by the quinol oxidation state and that both RssB and ArcA influence the abundance of the sigma factor σ³⁸ (RpoS) that in turn influences the expression of biofilm components in *E. coli* (71, 75–78). To determine whether each bladder defective TCS

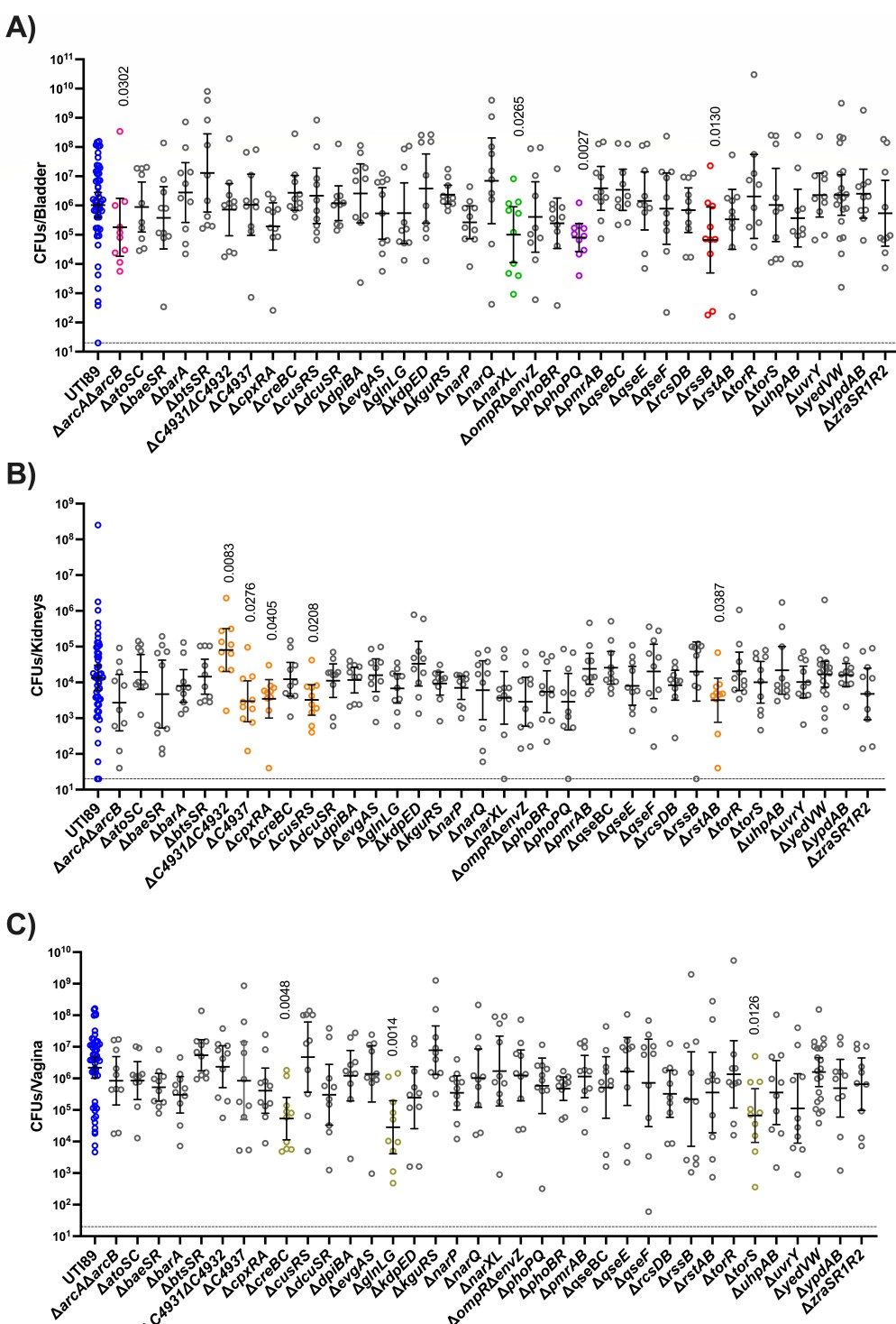

**FIG 3** Niche-specific contribution of TCSs during 24 h infection. Mice infected with UTI89 (blue) and isogenic TCS deletion strains were euthanized at 24 hpi for bacterial enumeration of titer within the (A) bladder, (B) kidneys, and (C) vagina. Each dot represents organ titers from a different mouse. The horizontal dotted line represents the limit of detection (20 CFUs). The solid line represents the geometric mean, and error bars depict the 95% confidence interval for each TCS. Additional colors correspond to TCSs in Fig. 4. A non-parametric Kruskal–Wallis with two-sided uncorrected Dunn's *post hoc* test was performed for statistical analysis.

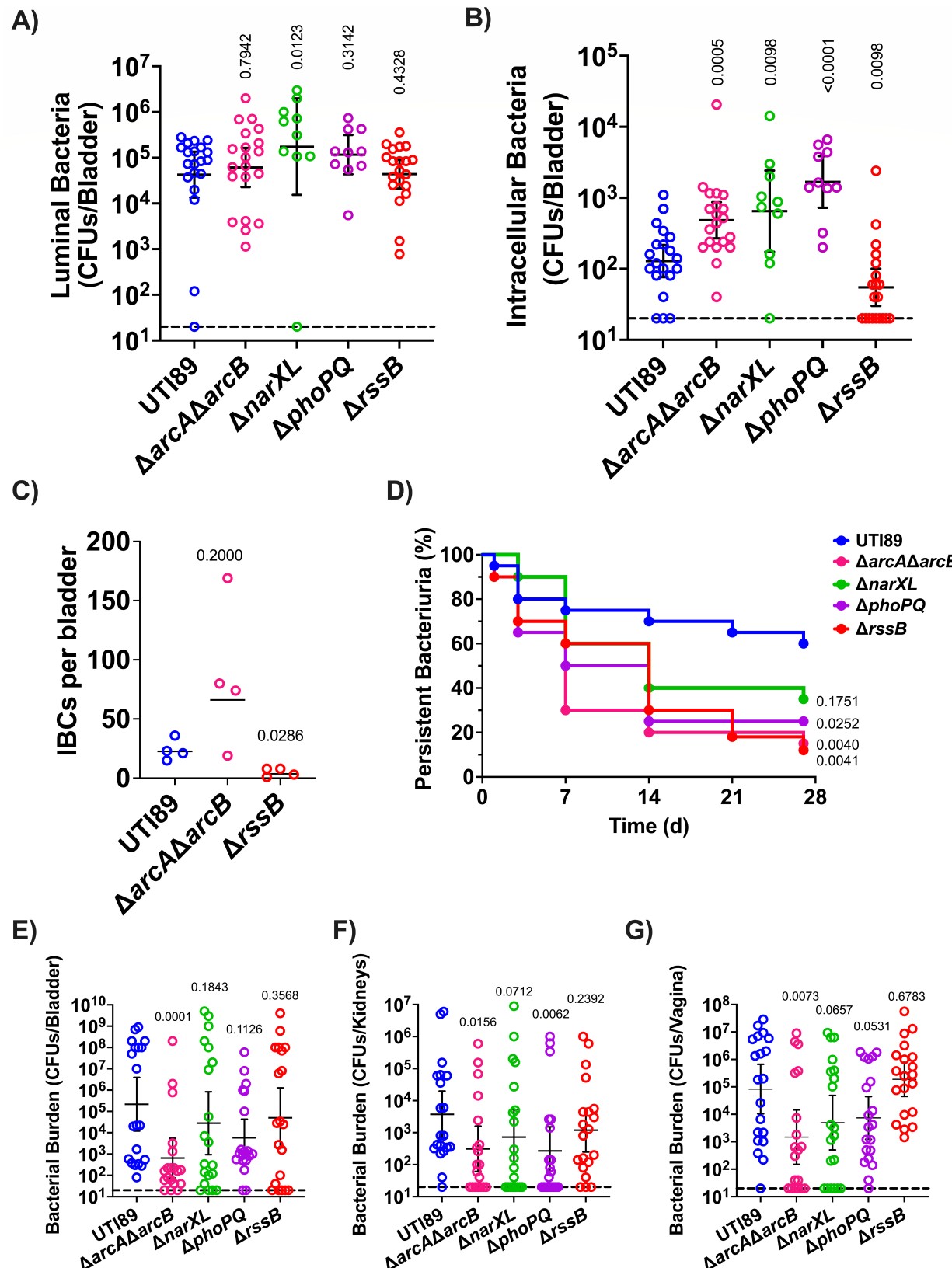

**FIG 4** TCSs contribute to distinct stages of extracellular and intracellular pathogenesis. (A and B) Graphs depict the (A) luminal and (B) intracellular bacterial titers from mouse bladders assessed at 6 hpi. Each symbol is a mouse. The horizontal dotted line represents the limit of detection (20 CFUs). The solid line represents the geometric mean, and error bars depict the 95% confidence interval of each TCS. Statistical analysis was performed with Mann–Whitney $U$ test. (Continued on next page)

**FIG 4** (Continued)

(C) Graph depicts the number of IBCs enumerated in each infected bladder at 6 hpi. Enumeration of IBCs was performed using confocal microscopy on randomly selected bladders. The line represents the geometric mean. Statistical analysis was performed with Mann–Whitney $U$ test. (D–G) Strains with aberrant acute infection titers were assessed in a long-term 28-day UTI model. (D) Graph depicts time-to-resolution curves defined as urine bacterial titers dropping below $10^4$ CFUs/mL. Vertical axis represents percentage of mice ($n = 20$) with persistent bacteriuria. Time to event was modeled with Kaplan–Meier method with non-parametric Mantel–Cox test for statistical comparison of TCS deletion mutants to UTI89. After 28-day urine analysis, bacterial titers were enumerated in the mouse (E) bladder, (F) kidneys, and (G) vagina. Each symbol is a mouse. The horizontal dotted line represents the limit of detection (20 CFUs). The solid line represents the geometric mean, and error bars depict the 95% confidence interval of each TCS. Statistical analysis was performed with Mann–Whitney $U$ test.

deletion mutant has the ability to form IBCs during acute infection, we evaluated intracellular bacterial titers and IBC formation at 6 hpi. To enumerate the extracellular and intracellular bladder populations, mice were euthanized at 6 hpi, and bladders were removed, bisected, and gentamicin-treated to eliminate extracellular bacteria and enable enumeration of the intracellular bacterial levels. These analyses revealed that all four mutants, Δ*arcA*Δ*arcB*, Δ*narXL*, Δ*phoPQ*, and Δ*rssB*, are retained in the bladder lumen with titers comparable to the wild-type parent (Fig. 4A); in contrast to the extracellular fractions, the intracellular titers of the mutants were statistically significantly different than the parental strain titers. The mutants Δ*arcA*Δ*arcB*, Δ*narXL*, and Δ*phoPQ* had high intracellular titers, whereas the Δ*rssB* mutant has drastically lower titers than the parental UTI89 strain (Fig. 4B) that are reminiscent of a mutant that lacks cytochrome bd (70). Based upon the 6 hpi bacterial titer analysis, we sought to determine whether the *rssB* or *arcA/arcB* deletion resulted in altered IBC morphology or numbers. We performed microscopy on these selected strains at 6 hpi. We did not observe any apparent difference in IBC morphology or size (Fig. S1). We noted an abundance of IBCs in the Δ*arcA*Δ*arcB* strain (Fig. 4C), which is consistent with the observed increase in intracellular bacterial titers. Conversely, there was a statistically significant lower number of IBCs in the Δ*rssB* infected bladders compared to the UTI89 strain (Fig. 4C), in agreement with the significantly lower numbers of intracellular numbers observed for the Δ*rssB* mutant.

Following acute infection, C3H/HeN mice may develop chronic cystitis, or the infection may resolve as indicated by bacterial urine titers decreasing below $10^4$ CFUs/mL (60). Chronic or resolved infection depends upon the bacteria clearing an early host-pathogen checkpoint within the first 24 h of acute infection (79, 80). We followed the urine titers of mice infected with either UTI89, Δ*arcA*Δ*arcB*, Δ*narXL*, or Δ*phoPQ* for 28 dpi and measured bacterial organ titers on day 28. Longitudinal urinalysis showed that despite harboring higher numbers of intracellular bacteria at the 6 h time-point, mice infected with Δ*arcA*Δ*arcB* had a shorter time to resolution, compared to wild-type UTI89 (Fig. 4D; Fig. S2). Similarly, urinalysis showed that the Δ*phoPQ*, and Δ*rssB* infected mouse cohorts had a shortened time to resolution compared to those infected with UTI89 (Fig. 4D). At the 28-dpi endpoint, the Δ*arcA*Δ*arcB* mutant was the only strain that was statistically significant in lower kidneys, bladder, and vagina titers than the parental UTI89 strain (Fig. 4E through G). The Δ*phoPQ* strain was significantly lower in the kidneys (Fig. 4F).

On a broad scale, these data indicate that TCSs make niche- and time-specific contributions to dynamic UPEC pathogenesis. On a finer scale, these data begin to indicate that UPEC may need to alternate between aerobic and anaerobic respiration states during different stages of infection. Collectively, our study elucidates the niche-specific TCS requirements of UPEC infection; knowledge that we believe will seed future research on better understanding how UPEC regulates virulence and fitness determinants in a dynamic manner.

## DISCUSSION

While studies have thus far extensively focused on UPEC virulence factors, very few reports have investigated regulatory pathways associated with UPEC pathogenic potential. Moreover, while extensive work has been performed at the molecular level on

TCSs biochemistry, only limited studies have evaluated the role of specific TCSs in UPEC pathogenesis (81). TCSs are major regulatory mechanisms that allow bacteria to switch between molecular tools in response to varying stimuli in their current environment. Here, we generated a deletion library of all the TCSs in the prototypical UPEC strain UTI89 as a tool to probe mechanisms of pathogenesis and persistence. We hope the future use of this library in the field enhances research in this area.

Our study highlights two critical aspects of TCSs in UPEC pathogenesis: (i) they are utilized under specific conditions and (ii) likely form complex networks. While there were no major differences between deletion strains during *in vitro* growth, clear molecular tissue tropisms were observed with *in vivo* experiments for a subset of TCS deletion strains. Previous studies have connected a few TCSs to UPEC pathogenesis with a targeted individual approach. For instance, deletion of the *ompR* RR in UPEC strain NU149, which impacts *fim* expression, resulted in approximately 2-log reduction in bladder and kidney titers in BACLB/c mice (82). Deletion of *cpxAR* diminished UTI89 fitness in the bladder after 3 dpi in CBA/J mice (83). Studies in UPEC strain CFT073 revealed that deletion of *barA-uvrY* attenuated bladder and kidney titers at 3 dpi, and that this attenuation was attributed to decreased LPS and hemolysin production (84). Deletion of *qseC* alone was documented to lead to attenuation of UPEC strain UTI89 in the mouse bladder (43). Notably, some of these prior studies investigated single-component deletions of either the RR or the HK, which can unmask non-partner interactions across TCSs (85–87). For this study, we generated TCS pair deletions for almost all TCSs, with the exception of orphan components like RssB, or systems in which the deletion of both TCS partners proved difficult to obtain (*barA/uvrY* and *torR/torS*). This constructed inventory of TCS deletion mutants now allows for an expansive assessment of the TCSs and their downstream regulons within a variety of settings.

For the purposes of the presented study, we focused on the acute phase of infection, where we found 12 different TCSs contributing to infection in either the bladder, kidney, or vaginal niches, focusing at 24 hpi (Fig. 3). The library can be leveraged further to evaluate the fitness of the TCS mutants at different stages of infection. Moreover, future studies can determine how the fitness potential of each TCS mutant changes if they are inoculated directly in the asymptomatic niches (vagina or gut) instead of transurethrally instilled in the bladder. Will different TCSs become important for exiting the asymptomatic reservoirs and ascending the urethra to the bladder? We think so.

We acknowledge that, in this study, we provide a broad view of a single cystitis isolate: the prototypical strain UTI89. UPEC genomic heterogeneity is extensive, and we posit that changes in genomic content may influence TCS use in a strain-specific fashion. Yet, we support that this study is significant as it provides the first comprehensive overview of TCSs to UPEC pathogenesis and lays the foundation for future in-depth investigations in other UPEC isolates and a comparison tool that is more representative than the model laboratory K-12 strains.

In the current study, and based on the expertise of our group, we selected to focus more closely with those TCS mutants that displayed a significant colonization defect in the bladder: Δ*arcA*Δ*arcB*, Δ*narXL*, Δ*phoPQ*, and Δ*rssB* (Fig. 4). Three of these mutants, Δ*arcA*Δ*arcB*, Δ*rssB*, and Δ*narXL* are convergent on the regulation of respiration, a critical aspect of UPEC pathogenesis. Under microaerobic conditions, ArcA is connected to the upregulation of *cydAB* and downregulation of *cyoABCDE* (88, 89). Recently, *cydAB*, which encodes the cytochrome *bd* oxidase, was found critical for proper IBC expansion (70). The expressions of *cydAB* and *arcA* are also regulated by the global one-component regulator FNR, which is active at low oxygen levels (88, 90). Along with FNR, the HK NarX, which helps to discern between nitrate and nitrite, phosphorylates NarL which in turn regulates *narG* expression, which is a nitrate reductase. The cytochrome *bd* oxidase and nitrate reductase renew the ubiquinone:ubiquinol pool though be it under different conditions (72). The HK ArcB controls the phosphorylated state of the RRs ArcA and RssB that co-regulate the balance in RpoS abundance in response to general stress like carbon starvation (71). While our results indicate that the TCS important to cystitis seems

to converge on respiration, the details to their direct or indirect interactions down or upstream of one another remain to be explored. Further study may reveal details of interconnectivity of these TCSs involved in an energetics balancing act.

In sum, two-component systems are critical systems that mediate a pathogen's ability to adapt behavior in response to external stressors. TCSs have global impacts on metabolism and virulence factors or targeted toward a narrow regulon. This work provides a comprehensive tool to dissect UPEC pathogenesis from a regulation standpoint and highlights that TCSs are important for UPEC pathogenesis in a niche-specific manner.

## ACKNOWLEDGMENTS

This work was supported by National Institutes of Health (NIH) grants T32GM007569 (J.R.B.), R01AI107052 (M.H.), P20DK123967 (M.H.), R01AI168468 (M.H.), F30AI169748 (S.A.R.), F30AI150077 (C.J.B.), and T32GM007347 (C.J.B.). Confocal laser scanning microscopy in Fig. S1 was performed at the Vanderbilt Cell Imaging Shared Resource (CISR), which is supported by NIH grant DK20593. Some images were created using BioRender.com.

J.R.B. conceived the study, performed most experiments, and composed the manuscript. S.A.R., T.A.B., and M.A.W. acquired and analyzed animal data in a blinded fashion. C.J.B. acquired and analyzed all confocal images in Fig. S1 (blinded). S.C. conducted growth curves in urine and growth curve analysis. T.L.R., T.L.D., and A.B. aided in the generation of the two-component system library. M.H. conceived the study and oversaw all aspects of its execution. All authors contributed to the generation, analysis, or interpretation of the data and edited the manuscript.

## AUTHOR AFFILIATIONS

[1]Department of Pathology, Microbiology and Immunology, Division of Molecular Pathogenesis, Vanderbilt University Medical Center, Nashville, Tennessee, USA
[2]Department of Urology, Vanderbilt University Medical Center, Nashville, Tennessee, USA

## PRESENT ADDRESS

John R. Brannon, Syneos Health, Morrisville, North Carolina, USA
Tamia Ross, Emory University School of Medicine, Atlanta, Georgia, USA
Adebisi Bamidele, Medical College of Georgia, Augusta, Georgia, USA

## AUTHOR ORCIDs

Seth A. Reasoner http://orcid.org/0000-0002-7791-1641
Sarah L. Comer http://orcid.org/0000-0003-2530-9840
Maria Hadjifrangiskou http://orcid.org/0000-0003-4249-8997

## FUNDING

| Funder | Grant(s) | Author(s) |
|---|---|---|
| HHS \| NIH \| National Institute of General Medical Sciences (NIGMS) | T32GM007569 | John R. Brannon |
| HHS \| NIH \| National Institute of Allergy and Infectious Diseases (NIAID) | R01AI107052, R01AI168468 | Maria Hadjifrangiskou |
| HHS \| NIH \| National Institute of Diabetes and Digestive and Kidney Diseases (NIDDK) | P20DK123967 | Maria Hadjifrangiskou |
| HHS \| NIH \| National Institute of Allergy and Infectious Diseases (NIAID) | F30AI169748, F30AI150077 | Seth A. Reasoner<br>Connor J. Beebout |

| Funder | Grant(s) | Author(s) |
|---|---|---|
| HHS | NIH | National Institute of General Medical Sciences (NIGMS) | T32GM007347 | Connor J. Beebout |

## AUTHOR CONTRIBUTIONS

John R. Brannon, Conceptualization, Data curation, Formal analysis, Validation, Writing – original draft, Writing – review and editing | Seth A. Reasoner, Formal analysis, Investigation, Methodology, Writing – review and editing | Tomas A. Bermudez, Formal analysis, Methodology, Validation, Writing – review and editing | Sarah L. Comer, Formal analysis, Methodology, Writing – review and editing | Michelle A. Wiebe, Formal analysis, Investigation, Methodology, Validation, Writing – review and editing | Taryn L. Dunigan, Methodology, Validation, Writing – review and editing | Connor J. Beebout, Formal analysis, Methodology, Validation, Visualization, Writing – review and editing | Tamia Ross, Data curation, Methodology, Writing – review and editing | Adebisi Bamidele, Investigation, Methodology, Validation, Writing – review and editing | Maria Hadjifrangiskou, Conceptualization, Data curation, Formal analysis, Funding acquisition, Investigation, Project administration, Software, Supervision, Validation, Visualization, Writing – original draft, Writing – review and editing

## ETHICS APPROVAL

Ethics approval for mouse experiments was granted by the VUMC Institutional Animal Care and Use Committee (IACUC) (protocol number M1800101-01). Experiments were performed in accordance with the guidelines of the National Institute of Health and IACUC at VUMC.

## ADDITIONAL FILES

The following material is available online.

### Supplemental Material

**Source Data File S1 (Spectrum02236-23-s0001.xlsx).** Raw data associated with Fig. 1 and S1.
**Source Data File S2 (Spectrum02236-23-s0002.xlsx).** Raw data associated with Fig. 2, 3, 4, and S3.
**Supplemental figures (Spectrum02236-23-s0003.pdf).** Figures S1 and S2.
**Supplemental tables (Spectrum02236-23-s0004.xlsx).** Tables S1 and S2.

### Open Peer Review

**PEER REVIEW HISTORY (review-history.pdf).** An accounting of the reviewer comments and feedback.

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
