## [Reviewer comments · Microbiology Spectrum]

Microbiology Spectrum

Mapping Niche-specific Two-Component System Requirements in Uropathogenic *Escherichia coli*

John Brannon, Seth Reasoner, Tomas Bermudez, Sarah Comer, Michelle Wiebe, Taryn Dunigan, Connor Beebout, Tamia Ross, Adebisi Bamidele, and Maria Hadjifrangiskou

Corresponding Author(s): Maria Hadjifrangiskou, Vanderbilt University Medical Center

Review Timeline:

Submission Date:	May 26, 2023
Editorial Decision:	July 7, 2023
Revision Received:	January 17, 2024
Accepted:	January 19, 2024

Editor: Philip Rather

Reviewer(s): The reviewers have opted to remain anonymous.

Transaction Report:

DOI: <https://doi.org/10.1128/spectrum.02236-23>

July 7, 2023

Dr. Maria Hadjifrangiskou
Vanderbilt University Medical Center
Pathology, Microbiology & Immunology, Division of Molecular Pathogenesis; Department of Urologic Surgery
1161 21st Avenue S
A5225A MCN
Nashville 37232

Re: Spectrum02236-23 (Mapping Niche-specific Two-Component System Requirements in Uropathogenic Escherichia coli)

Dear Dr. Hadjifrangiskou,

Thank you for submitting your manuscript to Microbiology Spectrum. Your manuscript has been reviewed by two experts in the field and both were supportive of the work, but have comments that will need to be addressed. In particular, note the comments from each reviewer regarding the growth curve analysis.

Link Not Available

Sincerely,

Philip Rather

Journals Department
Reviewer comments:

Reviewer #1 (Comments for the Author):

In this manuscript, Brannon et al identified and disrupted 32 TCSs in uropathogenic E. coli strain UTI89 and assessed contribution to in vitro growth, bladder epithelial cell adhesion and invasion, and colonization in a murine model of UTI. None of the mutants exhibited in vitro defects, but specific differences were observed in vivo in the bladder, kidneys, and vagina, suggesting niche-specific roles for TCSs. The manuscript is well written and provides interesting new information regarding the contribution of TCSs to UPEC pathogenesis. Main suggestions are as follows:

I found it surprising that none of the TCS mutants exhibited growth differences in vitro, particularly in minimal medium. Where there difference in density at which they achieved stationary phase? Have growth differences been found for any of these systems in other E. coli or Enterobacteriaceae? It would be helpful to include this information as part of the discussion.

The manuscript text appropriately qualifies the results by saying no defects were detected under the tested experimental conditions, and I do not feel that further testing of the full panel of mutants is necessary. However, the results would be strengthened by testing the mutants with in vivo bladder defects for growth in human urine.

The y-axis label of Figure 3D does not appear to correspond to the presented data. It would also be helpful to show the temporal urine CFU data as a supplemental figure.

Reviewer #2 (Comments for the Author):

This paper describes the construction of a complete set of two-component system knockouts in the urinary pathogenic E. coli UTI89. The various knockouts are characterized for growth in vitro, adherence to or invasion of urothelial cells, and colonization of the bladder, kidneys and vagina in a mouse UTI model. The authors find that different TCSs are important for colonization of different niches. The results highlight some systems associated with respiration. I believe the results will be of interest to the community studying UTIs and also to those interested more generally in host-microbe interactions and two-component signaling. In addition, the strain collection will be a great resource for this community.

I have only a few specific comments.

1) Lines 217-218 and Fig. 1. The results indicate that UTI89 has a specific growth rate of about 0.2/hr in both LB and N-minimal medium. This result raises two questions: a) Why does UTI89 grow so slowly in LB at 37 degrees with shaking? A specific growth rate of 0.2/hr is a doubling time of about 3.5 hours, which is much longer than the behavior of most E. coli strains in LB; b) How is it possible that UTI89 has virtually the same growth rate in LB and a minimal medium? Is this N-minimal medium in fact a very rich medium? If so, then it should not be described as a minimal medium. These questions raise concerns that there was a problem with the growth measurements or data analysis, so they need to be addressed.

2) Related to the above comments, both a reference for N-minimal medium and the recipe, including carbon source and supplements, if any, should be in the methods section of the paper.

3) There are a number of typos in Table 1 ("Presence", Changes is osmolarity", ...). Also, line 230 mentions kguRS but I do not see the names KguR or KguS in Table 1.

4) Lines 257-261. DtorS showed significant colonization defects in Fig. 2C, but deletion of its partner response regulator, DtorR, did not show such a defect. Doesn't this run counter to the conclusion that the TorRS TCS contributes to survival within the niche (line 260)? Perhaps deleting torS leads to inappropriate activation of TorR or possibly TorS also phosphorylates another response regulator. This should be discussed or at least the summary conclusion in line 260 "these TCSs contribute to survival within a niche" should be adjusted accordingly.

5) Fig. 2 I found it confusing that the deletions were not in the same order in panels A, B, C. I suggest keeping the same order across all three panels and using colors to highlight the significant systems in the various panels.

Staff Comments:

Preparing Revision Guidelines

For complete guidelines on revision requirements, please see the journal Submission and Review Process requirements at

<https://journals.asm.org/journal/Spectrum/submission-review-process>. **Submissions of a paper that does not conform to Microbiology Spectrum guidelines will delay acceptance of your manuscript. "**

Please return the manuscript within 60 days; if you cannot complete the modification within this time period, please contact me. If you do not wish to modify the manuscript and prefer to submit it to another journal, please notify me of your decision immediately so that the manuscript may be formally withdrawn from consideration by Microbiology Spectrum.

VANDERBILT UNIVERSITY

School of Medicine

Maria Hadjifrangiskou, Ph.D.
Associate Professor
Department of Pathology, Microbiology, and Immunology
Vanderbilt University School of Medicine
AA-4210 Medical Center North
1161 21st Avenue South
Nashville, TN 37232-8240
Phone: (615) 322-4851
Maria.hadjifrangiskou@vumc.org

Dear Professor Rather;

Thank you for considering our manuscript no. Spectrum02236-23, titled "Mapping Niche-specific Two-Component System Requirements in Uropathogenic *Escherichia coli*" for publication in *Microbiology Spectrum*, as well as the opportunity to submit a revised manuscript for further consideration.

We were pleased to read that each expert reviewer had favorable impressions of our manuscript, commenting that our study provides new insights into the colonization requirements of UPEC, and we appreciate the insightful feedback provided. In response to suggestions by reviewer 1 we now provide new data, demonstrating *in vitro* growth curves in urine. Because the first author of the study has since left the lab, another lab member performed the suggested experiments and subsequent data analysis. This resulted in my adding the said researcher, Miss Sarah Comer, on the author list. A point-by-point response to each reviewer's comments can be found in this document.

Reviewer Comments:

REVIEWER #1:

In this manuscript, Brannon et al identified and disrupted 32 TCSs in uropathogenic E. coli strain UT189 and assessed contribution to in vitro growth, bladder epithelial cell adhesion and invasion, and colonization in a murine model of UTI. None of the mutants exhibited in vitro defects, but specific differences were observed in vivo in the bladder, kidneys, and vagina, suggesting niche-specific roles for TCSs. The manuscript is well written and provides interesting new information regarding the contribution of TCSs to UPEC pathogenesis.

Response: We thank the reviewer for the positive review and helpful comments. Responses to the proposed suggestions are found below:

Reviewer: *I found it surprising that none of the TCS mutants exhibited growth differences in vitro, particularly in minimal medium. Were there differences in density at which they achieved stationary phase? Have growth differences been found for any of these systems in other E. coli or Enterobacteriaceae? It would be helpful to include this information as part of the discussion.*

Response: We thank the reviewer for this comment. We now provide more detailed data on the growth of different strains in N-minimal media versus LB. We need to note here that there was an error made during the calculation of the LB specific growth rate, which was pointed out by reviewer 2 and prompted us to repeat the growth curves and re-analyze our data. We now provide extended data on all strains, along with final OD600 at stationary phase in each condition tested. We also now provide growth in an infection-relevant medium, urine.

Reviewer: *The manuscript text appropriately qualifies the results by saying no defects were detected under the tested experimental conditions, and I do not feel that further testing of the full panel of mutants is necessary. However, the results would be strengthened by testing the mutants with in vivo bladder defects for growth in human urine.*

Response: We agree with reviewer 1 regarding the value of obtaining growth curves in pooled human urine, at least for those strains that display *in vivo* defects. We now include urine growth data for all strains in the revised manuscript.

Reviewer: *The y-axis label of Figure 3D does not appear to correspond to the presented data. It would also be helpful to show the temporal urine CFU data as a supplemental figure.*

Response: We have modified the y-axis label to read “Persistent Bacteriuria”. Figure 3D depicts a “time-to-resolution” of chronic bacteriuria. The vertical axis of Figure 3D represents the percent of mice (n=20 per group) with bacteriuria ($>10^4$ CFUs/mL) at each timepoint. Once a mouse’s urine bacterial titer drops below 10^4 CFUs/mL, that mouse is considered resolved of its chronic infection and is removed from the corresponding curve in Figure 3D regardless of whether the mouse later experiences high-titer bacteriuria. The stepwise nature of the graph is due to the days at which urinalysis was conducted (days 1, 3, 7, 14, 21, & 27). We have now included additional information in the legend and the methods section explaining the y-axis title. We also provide the bacteriuria data as supplementary figure S2.

Reviewer #2 Major comments:

Reviewer: *This paper describes the construction of a complete set of two-component system knockouts in the urinary pathogenic E. coli UTI89. The various knockouts are characterized for growth in vitro, adherence to or invasion of urothelial cells, and colonization of the bladder, kidneys and vagina in a mouse UTI model. The authors find that different TCSs are important for colonization of different niches. The results highlight some systems associated with respiration. I believe the results will be of interest to the community studying UTIs and also to those interested more generally in host-microbe interactions and two-component signaling. In addition, the strain collection will be a great resource for this community.*

Response: We thank the reviewer finding our work significant and well-done and appreciate the reviewer’s careful evaluation and helpful comments. Responses to the proposed suggestions are found below:

Reviewer: Lines 217-218 and Fig. 1. The results indicate that UTI89 has a specific growth rate of about 0.2/hr in both LB and N-minimal medium. This result raises two questions: a) Why does UTI89 grow so slowly in LB at 37 degrees with shaking? A specific growth rate of 0.2/hr is a doubling time of about 3.5 hours, which is much longer than the behavior of most *E. coli* strains in LB; b) How is it possible that UTI89 has virtually the same growth rate in LB and a minimal medium? Is this N-minimal medium in fact a very rich medium? If so, then it should not be described as a minimal medium. These questions raise concerns that there was a problem with the growth measurements or data analysis, so they need to be addressed.

Response: We thank the reviewer for catching this. There was a carryover error during the analysis of the data that resulted in this discrepancy. The reviewer is correct in that UTI89 should have a faster growth rate in LB and it reaches a higher final OD600. We now present the re-analysis of the specific growth rate. We also need to point out, as described in the materials and methods, that the growth curves were performed in a plate reader using small liquid volumes. While this is convenient for allowing the parallel comparison of the different strains, we admit that these conditions may not mimic growth in batch culture.

Reviewer: Related to the above comments, both a reference for N-minimal medium and the recipe, including carbon source and supplements, if any, should be in the methods section of the paper.

Response: We apologize for the omission. We now include a reference for the N-minimal media recipe.

Reviewer: There is a number of typos in Table 1 ("*Prescence*", *Changes is osmolarity*", ...). Also, line 230 mentions *kguRS* but I do not see the names *KguR* or *KguS* in Table 1.

Response: We thank the reviewer for noticing typographical errors in Table 1 and the inadvertent omission of *kguRS* within this table. We have edited Table 1 to correct these errors.

Reviewer: Lines 257-261. Δ *torS* showed significant colonization defects in Fig. 2C, but deletion of its partner response regulator, Δ *torR*, did not show such a defect. Doesn't this run counter to the conclusion that the *TorRS* TCS contributes to survival within the niche (line 260)? Perhaps deleting *torS* leads to inappropriate activation of *TorR* or possibly *TorS* also phosphorylates another response regulator. This should be discussed or at least the summary conclusion in line 260 "*these TCSs contribute to survival within a niche*" should be adjusted accordingly.

Response: The reviewer here raises an excellent point. We have made the relevant adjustment to the document.

Reviewer: Fig. 2 I found it confusing that the deletions were not in the same order in panels A, B, C. I suggest keeping the same order across all three panels and using colors to highlight the significant systems in the various panels.

Response: We thank the reviewer for this suggestion and have changed the graphs accordingly.

Re: Spectrum02236-23R1 (Mapping Niche-specific Two-Component System Requirements in Uropathogenic Escherichia coli)

Dear Dr. Hadjifrangiskou:

Your manuscript has been accepted, and I am forwarding it to the ASM production staff for publication. Your paper will first be checked to make sure all elements meet the technical requirements. ASM staff will contact you if anything needs to be revised before copyediting and production can begin. Otherwise, you will be notified when your proofs are ready to be viewed.

Sincerely,
Philip Rather
Editor
Microbiology Spectrum